# Behavioural Responses of Defended and Undefended Prey to Their Predator—A Case Study of Rotifera

**DOI:** 10.3390/biology11081217

**Published:** 2022-08-13

**Authors:** Victor Parry, Ulrike E. Schlägel, Ralph Tiedemann, Guntram Weithoff

**Affiliations:** 1Unit of Ecology and Ecosystem Modelling, University of Potsdam, 14469 Potsdam, Germany; 2Unit of Plant Ecology and Nature Conservation, University of Potsdam, 14476 Potsdam, Germany; 3Unit of Evolutionary Biology/Systematic Zoology, University of Potsdam, 14476 Potsdam, Germany; 4Berlin-Brandenburg Institute of Advanced Biodiversity Research, 14195 Berlin, Germany

**Keywords:** animal behaviour, transgenerational response, *Brachionus calyciflorus*, *Asplanchna brightwellii*, video analysis

## Abstract

**Simple Summary:**

Many animals that have to cope with predation have evolved mechanisms to reduce their predation risk. One of these mechanisms is change in morphology, for example, the development of spines. These spines are induced, when mothers receive chemical signals of a predator (kairomones) and their daughters are then equipped with defensive spines. We studied the behaviour of a prey and its predator when the prey is either defended or undefended. We used common aquatic micro-invertebrates, the rotifers *Brachionus calyciflorus* (prey) and *Asplanchna brightwellii* (predator) as experimental animals. We found that undefended prey increased its swimming speed in the presence of the predator. The striking result was that the defended prey did not respond to the predator’s presence. This suggests that defended prey has a different response behaviour to a predator than undefended conspecifics. Our study provides further insights into complex zooplankton predator-prey interactions.

**Abstract:**

Predation is a strong species interaction causing severe harm or death to prey. Thus, prey species have evolved various defence strategies to minimize predation risk, which may be immediate (e.g., a change in behaviour) or transgenerational (morphological defence structures). We studied the behaviour of two strains of a rotiferan prey (*Brachionus calyciflorus*) that differ in their ability to develop morphological defences in response to their predator *Asplanchna brightwellii*. Using video analysis, we tested: (a) if two strains differ in their response to predator presence and predator cues when both are undefended; (b) whether defended individuals respond to live predators or their cues; and (c) if the morphological defence (large spines) *per se* has an effect on the swimming behaviour. We found a clear increase in swimming speed for both undefended strains in predator presence. However, the defended specimens responded neither to the predator presence nor to their cues, showing that they behave indifferently to their predator when they are defended. We did not detect an effect of the spines on the swimming behaviour. Our study demonstrates a complex plastic behaviour of the prey, not only in the presence of their predator, but also with respect to their defence status.

## 1. Introduction

Trophic interactions are the most direct interactions between organisms, often causing severe harm or death to the prey. Hence, predation serves as a strong selective force in communities. Highly vulnerable organisms have evolved mechanisms to reduce predation risks [1]. These include the ability to avoid detection by their predator (camouflage and crypsis), efficient detection of approaching predators, escape responses and antipredator morphological defences, which may be either permanent or induced by chemical cues released by the predators [2,3,4,5,6]. In aquatic communities, zooplankton are greatly affected by vertebrate and invertebrate predators. Zooplankton species can sense their predators directly or via chemical cues that may induce behavioural and/or morphological defences [2,7,8,9,10,11,12]. Induced behavioural responses lower the probability of detection or, upon detection, increase the chance of escape [13]. They include the diel vertical movement of crustaceans [14], change in swimming speed due to predator presence [1], escape or evasive behaviour [9,15] and floating behaviour [12]. Morphological defences increase the probability of surviving an attack. Defensive structures increase the overall body size, thereby increasing the handling time for predators [16] or protecting the prey from being ingested. Such transgenerationally (i.e., parthenogenetic mothers perceive the predator presence and their daughters are defended) induced responses have been shown to be very efficient in surviving a predator’s attack [17,18,19,20,21].

Rotifers are cosmopolitan micro-invertebrates that play an integral role in planktonic food webs, and are prey to various predators. A number of species have developed strategies to survive predation by induced morphological defences i.e., increasing spine length [22,23,24,25,26,27] and/or behavioural strategies [1,8,15]. Most studies on the response of rotifers to predation risk have focused on behavioural responses before the induction of transgenerational morphological responses and neglected the behavioural response to predators after induction of morphological structures. It is unclear if morphologically defended preys also respond behaviourally to predator presence and predator cues. We aim to investigate the behavioural response of a prey to its predator with and without morphological defence. An effective morphological defence would increase the chances of prey survival even when attacked, hence, possibly making a behavioural response superfluous.

Therefore, we studied a common predator–prey pair of the two rotifer species: *Brachionus calyciflorus* (prey) and *Asplanchna brightwellii* (predator). *A. brightwellii* is an efficient predator to many rotifer species (Seifert et al., [28]) and recognizes its prey using chemo- and mechanoreceptors [11,23,29,30]. Thus, encounters between predator and prey are mainly driven by swimming behaviour.

We analysed the behavioural responses (changes in swimming speed and directional persistence) of predator and prey using video analysis [31]. We first compared the behavioural response of two strains of undefended prey specimens, one with the ability to grow spines and the other one without, to live predator or the predator’s cues. Secondly, we analysed the behavioural response of the spined prey. We aim to test the following hypotheses: (a) both predator and prey respond behaviourally to the presence of their prey/predator; (b) defended prey exhibits a different response to a predator compared with undefended prey; and (c) the degree of defence, expressed as spine length has an effect on swimming behaviour.

## 2. Materials and Methods

### 2.1. Study Organisms

The predator *Asplanchna brightwellii* was originally isolated from a small, shallow, urban pond in a park area (Im Schwarzen Grund) in Berlin (Germany, 52°29′ N, 13°17′ E) that is surrounded by a reed belt and grass meadows. *Asplanchna* was reared in stock cultures with two strains of *Brachionus calyciflorus* sensu stricto [32] (strain IGB and Michigan, see below) and *Keratella cochlearis* as food. This food mixture has been proven to promote stable cultures, whereas a monospecific diet often leads to unstable boom–bust dynamics. For the experiment on the response of unspined prey to predation, we used the strains “IGB” and “Michigan” from cultures that were regularly diluted to keep the population in the exponential growth phase. Strain “IGB” exhibits only a slight morphological response with almost no spine growth. This fact makes it more likely to show an immediate behavioural response, whereas the strain “Michigan” exhibits a prominent morphological response growing long spines. A 50:50 mixture of the coccal chlorophyte *Monoraphidium minutum* (SAG Culture Collection, Göttingen, Germany, strain number 243-1) and the flagellated cryptophyte *Cryptomonas* sp. (SAG, strain number 26-80) served as food for both *Brachionus* strains.

For the experiment on the transgenerational response, we used the strain “Michigan”, known to exhibit large spine induction. Therefore, *B. calyciflorus s.s.* strain “Michigan” was cultured together with its predator *A. brightwellii* (body size ranged from 500 to 650 µm) and *B. urceolaris* as combined food for the predator for 2 to 3 weeks i.e., several generations of predator and prey. The undefended *B. urceolaris* served as a valuable food for the growth of *A. brightwellii*, which in turn induced a pronounced but variable spine production resulting in the co-occurrence of animals with different spine lengths. All animals reproduced exclusively asexually in these stock cultures, thus, only amictic females were used in our experiments. All animals and algae were cultured at 20 °C with a light:dark cycle of 16:8 h in a modified WC medium [33].

### 2.2. Video Tracking Setup and Settings

For video tracking of the animals we used a Canon 6D camera, Canon MP-65 macro lens, a focusing micrometric slide, a stereomicroscope base and a laptop for recording the videos. Recordings were performed with 25 frames per second (fps), f:/9 aperture, ISO 200, and 1/30 s of exposure time. The only source of light was provided by stereomicroscope white light. We recorded videos of a length of 30 s to analyse the swimming behaviour, in particular the swimming speed and the directional persistence [34]. All recordings were performed under the same light conditions.

### 2.3. Experimental Design

#### 2.3.1. Predator and Prey Behaviour with Unspined Prey

In the experiments, the size of *B. calyciflorus* ranged from 100 to 200 µm and the size of *A. brightwellii* ranged from 500 to 650 µm. Experiments were designed to examine the behavioural response of unspined *B. calyciflorus* to the different environments of predation. We applied three treatments for both *B. calyciflorus* strains: a control where *B. calyciflorus* was filmed in WC medium without exposure to predators, a treatment where *B. calyciflorus* was exposed to *A. brightwellii* and a treatment where *B. calyciflorus* was exposed to predator cues. For all treatments, 15 parallels were setup and filming started after one hour of acclimation for 90 s (three 30 s length videos). For all treatments, five non-egg-bearing *B. calyciflorus* individuals were randomly chosen and placed in wells of a 12-well microtiter plate. These wells had an area of 3.9 cm^2^ and a diameter of 22 mm, which is 110 to 220 times the length of the prey and were larger than the ones used in similar studies [1,35]. These arenas, in principle, allow for three-dimensional movement, however, most of the movement took place in a two-dimensional plane. To test for a potential crowding effect, we ran an initial experiment with one, five, eight and 20 animals per well and we found no differences in swimming speed among different animal densities.

For the control, the individuals were transferred with a glass pipette from the stock culture into a well, filled with 1 mL fresh WC medium. 

For tests where *B. calyciflorus* was exposed to the presence of live *A. brightwellii*, we starved the *Asplanchna* culture for 12 h prior to conducting the experiments. One individual *A. brightwellii* was transferred into a 12-well microtiter plate filled with 1 mL fresh WC medium and filmed after an hour as the control treatment for *A. brightwellii*. Afterwards, five *B. calyciflorus* individuals were transferred from stock culture into the well with one individual of *A. brightwellii*. The response of *B. calyciflorus* to *A. brightwellii* was recorded after an hour of acclimation. During this acclimation period, *A. brightwellii* had already eaten one or more prey individuals in some treatments. The number of eaten animals had neither an effect on the mean swimming speed nor on the directional persistence of the prey (see Appendix A).

To test for effects of predator cues on *B. calyciflorus*, they were exposed to culture medium from a dense *A. brightwellii* culture with a density of about 3 ind mL^−1^. Prior to its use in the experiment, the medium was carefully sieved through 30 µm mesh to remove all predators but keeping potential kairomones in the medium.

#### 2.3.2. Predator and Prey Behaviour with Spined Prey (Transgenerational)

##### Predator and Prey Behaviour

To test whether spined *B. calyciflorus* also responds to the presence of live *A. brightwellii*, we analysed the swimming behaviour of the same spined individual in the presence and absence of the predator in a “one prey one predator” ratio in a 12-well microtiter plate as above. Prior to the experiment, non-egg-bearing *Brachionus* individuals were taken randomly from stock cultures and kept for 30 min in a petri dish to remove potential predator cues. Then, both predator and prey individuals were acclimated for one hour in separate wells and filmed as a control. They were then placed together in the same well and after one hour of acclimation, their behaviour was filmed. In none of the cases was a spined prey individual eaten by *A. brightwellii*. After filming, the animals were fixed with Lugol´s iodine and we measured the spine length and body length using a video-aided inverted microscope (TSO, Thalheim, Germany). We analysed the swimming behaviour of predator and prey by comparing the swimming speed and directional persistence of both species alone and with its prey/predator. For all treatments, 13 wells were filmed and recorded for 90 s (three 30 s length videos).

##### Predator Cues (Kairomones) Treatment

For testing a potential kairomone (predator cue) effect, a similar set-up as above was used: Prior to the experiment, non-egg-bearing *Brachionus* individuals were taken randomly from stock cultures and kept for 30 min in a petri dish to remove potential predator cues. Then, the same individual animals were transferred into a well of 12-well microtiter plate filled with 1 mL fresh WC medium and were filmed after an hour as a control. After that, *Brachionus* was exposed to the kairomone for one hour by adding sieved, pre-conditioned medium from an *Asplanchna* culture, and filmed. The spine and body lengths were measured as above. For all treatments, 13 wells were filmed and recorded for 90 s (three 30 s length videos).

### 2.4. Video Analysis and Calculation of Swimming Speed and Directional Persistence

The movement of the organisms was tracked and the trajectories extracted using the BEMOVI package [31] of the R environment [36] and Image J (image analysis, Eliceiri et al. [37]). The raw videos were converted from *.MOV (file extension) to *.avi (file extension) format using the open-source software FFmpeg [38], which is required by Image J. To facilitate the analysis, we removed static parts or noise (i.e., dust) in the videos using the Image J process _*noise*_*despeckle*. We followed the analytical steps of trajectory extraction and the workflow described by Pennekamp et al. [31]. BEMOVI identifies and tracks the actual movement of individuals (based on morphology, abundance or behaviour) in videos. From these, movement characteristics such as movement speed, turning angles and step lengths are computed [31,34,39]. Mean swimming speed was calculated as the step speed (μm s^−1^) of trajectories extracted from the BEMOVI package. Step speed was computed as “step length” (based on a fixed time interval) divided by length of that time interval. Relative swimming speed was calculated as (μm s^−1^/body length) to account for differently sized animals (different spine lengths). For persistence, turning angles of trajectories were extracted from BEMOVI angular turns (“rel_angle”) and fitted with a wrapped Cauchy distribution using the *circular* package [40] to estimate directional persistence, which specifies how strongly turning angles are centred around zero. Directional persistence scales from zero to one with values close to 1 indicating that an individual is highly likely to move in the same direction as during the previous time step. For swimming speed and directional persistence, the mean speed and mean persistence of the prey were calculated from all prey individuals per well. The analyses were performed on an Intel Core^TM^ i7-4790 CPU @ 3.60 GHz, 32 GB RAM, x 64-based processor: GPU AMD Radeon R5 430.

### 2.5. Statistical Analysis

The recording and subsequent automated analysis of the data did not distinguish between the predator and the prey in the combined treatment. Thus, after the video analysis, trajectories of the individual animals were obtained without species assignment. To assign these trajectories to either *B. calyciflorus* or *A. brightwellii*, we used a random forest approach (supervised machine learning), which is a widely used classification algorithm [41]. To train the random forest, the morphological characteristics of *B. calyciflorus* and *A. brightwellii* from the single species treatments were used as templates. Parameters were perimeter, area, aspect ratio and speed as suggested by Pennekamp et al. [31] and Obertegger et al. [39]. The area, perimeter and speed best classified species according to the Gini importance index, with 2% misclassification error estimated by the out-of-bag error rate.

For immediate behavioural responses of unspined prey and their predator, we applied a multivariate test to determine the differences in speed and persistence amongst the different experimental groups. Since in the presence of the predator some prey individuals were eaten during the acclimation period, we included survival as a covariate. We applied analysis of variance (MANCOVA) using the *jmv* package [42] to test the hypothesis, (a), of significant differences in speed among groups and additionally used Tukey’s post hoc tests using the *car* package [43] for pairwise comparisons among experimental groups.

For transgenerational behavioural responses of spined prey and their predator, we calculated the relative spine length as the spine length divided by the body length. We applied linear regression models using the *lm* function of the *stats* package [36] to determine the effect of treatment, body length and spine length on persistence, swimming speed and relative swimming speed. We also calculated type II analysis-of-variance using the Anova function of the *car* package [43] to determine the differences between models of the treatments using body length and spine length as covariables. We also applied paired *t*-tests for comparison between treatments, as the same individuals were tested (non-independence). All analyses and calculations were performed using the R language and environment [36].

## 3. Results

### 3.1. Behavioural Responses with Unspined Prey

#### 3.1.1. Prey Behaviour

For *B. calyciflorus* strain “IGB”, the mean swimming speed in the control, without predators or predator cues, was 470 µm s^−1^ (±69, SD). Prey behaviour was significantly different among treatments (ANOVA, F = 11.06, df = 2, *p*-value < 0.001): in the presence of the predator, the swimming speed was ca. 30% higher compared with the control, whereas the speed in the kairomone treatment was not different from the control (Figure 1). For *B. calyciflorus* strain “Michigan”, there was also a significant effect of the treatment (ANOVA, F = 15.57, df = 2, *p*-value < 0.001): the mean swimming speed of *B. calyciflorus* in the control was 430 µm s^−1^ (±94, SD) and increased similarly in the presence of the predator, by 21% to 524 μm s^−1^ (±129, SD; Figure 1). This increase in swimming speed was reflected in a higher proportion of faster movements than in the control but not in faster maximum speed (Appendix A). When *B. calyciflorus* strain “Michigan” was exposed only to the predator cues (kairomones) of *A. brightwellii*, we observed a significant decrease in swimming speed to 319 μm s^−1^ (±65, SD). This decrease in swimming speed was associated with a marginal decrease in persistence (F = 3.18, df = 2, *p*-value = 0.052). For strain “IGB”, we found a significantly lower persistence in the predator cues (kairomone) treatment compared with the other two treatments (F = 4.68, df = 2, *p*-value = 0.011). This means more twists and turns than in the other treatments.

#### 3.1.2. Predator Behaviour

There was no significant difference in the mean swimming speed of *A. brightwellii* with or without prey for both *B. calyciflorus* strains: strain “IGB” (t = 1.04, df = 26.42, *p*-value = 0.308) (Figure 1) and strain “Michigan” (t = 1.26, df = 20.85, *p*-value = 0.223) (Figure 1). There was also no significant difference in the mean persistence of *A. brightwellii* with or without prey for both *B. calyciflorus* strains: strain “IGB” (t = 0.22, df = 26.91, *p*-value= 0.827) (Figure 1) and strain “Michigan” (t = 1.20, df = 20.57, *p*-value = 0.243) (Figure 1).

### 3.2. Transgenerational Behavioural Responses with Spine Prey

#### 3.2.1. Prey Behaviour

##### Live Predator Treatment

We analysed 13 animals (spined *B. calyciflorus* strain “Michigan”) of different spine lengths ranging from 65 to 226 µm. We found no differences in swimming speed, relative swimming speed and persistence for the defended individuals in the presence of their predator and the control (Table 1, Table 2 and Table 3; Appendix A). Using linear regression, we found that mean swimming speed decreased with body length for both control (df = 11, F= 6.94, r^2^ = 0.39, *p*-value= 0.023) and live predator treatment (df = 11, F= 5.64, r^2^ = 0.34, *p*-value= 0.037) (Appendix A). Body length had a negative effect on swimming speed and relative swimming speed (in relation to body length) for both treatments (Figure 2; Appendix A). Using ANCOVA, we found that body length as an independent variable had an effect on relative swimming speed (df = 1, F = 31.53, *p*-value < 0.001). Spine length as an independent variable alone had no significant effect, however, interaction with body length had a significant effect on relative swimming speed (df = 1, F = 7.49, *p*-value = 0.014; Table 2). Body length (df = 1, F= 12.38, *p*-value = 0.002) had an effect on directional persistence as an independent variable (Table 3).

##### Predator Cues (Kairomones) Treatment

We analysed 13 animals of different spine lengths ranging from 45 to 83 µm. We found no differences in swimming speed and relative swimming speed for the defended individuals in predator cues medium and control; however, we found significant difference in directional persistence between the treatments (ANCOVA, df = 1, F = 7.55, *p*-value= 0.013) (Table 1, Table 2 and Table 3; Appendix A). Linear regression analysis revealed that mean swimming speed decreased with body length in the control treatment (df = 11, r^2^ = 0.33, *p*-value= 0.041), however, we could not detect a difference in swimming speed with body length in the predator cues treatment (df = 11, r^2^ = 0.19, *p*-value = 0.125) (Figure 3; Appendix A). Using ANCOVA, body length had an effect on relative swimming speed as an independent variable (df = 1, F = 4.50, *p*-value= 0.048). The effects of the other independent variables (treatment and spine length) on swimming speed and relative swimming speed were significant only in interaction with body length. These numerous significant interaction terms demonstrate the complex interplay of the independent variables on the swimming behaviour (Table 1 and Table 2).

#### 3.2.2. Predator Behaviour

There was no significant difference in the mean swimming speed (paired *t*-test, df = 12, t = 0.63, *p* = 0.534) and persistence (paired *t*-test, df = 12, t = 0.39, *p*-value = 0.703) of *A. brightwellii* with or without spined prey for *B. calyciflorus* strain “Michigan (Appendix A).

## 4. Discussion

We used video-based analysis to study the behavioural responses of two strains of unspined *Brachionus calyciflorus* (“Michigan” and “IGB”) and one strain (“Michigan”) of spined prey to predation, by exposing them to live predator (*A. brightwellii*) or its kairomones (only predator cues). We found that behavioural responses of prey to predator depended on the environment (either with live predator or with only predator cues) and the induced morphological defence.

### 4.1. Behavioural Responses of Unspined Prey

We found an overall behavioural response with higher swimming speed for both unspined *Brachionus* strains in the presence of their predator. This is contrary to predictions and model simulations that indicate that prey reduces its swimming speed to minimize encounter rate with the predator [1,44]. The increase in swimming speed in our study might be attributed to the physical perception of the predator or its flow field [8,15,19,45]. An alternative explanation could be the prey´s need to optimize foraging even with predation risks after a period of acclimation. The overall effectiveness of behavioural defence depends on the density of predator (low predator number reduces prey–predator encounters) and feeding and reproduction needs. There is a trade-off between filter feeding and predation risk: a reduction in swimming speed in response to a predator leads to a reduced food intake for filter feeders that combine the action of swimming with feeding [1]. After 1 h of acclimation, prey may no longer be able to afford a reduction in speed in the presence of a predator as this can significantly affect feeding rates and fitness. Thus, they increase speed to enhance foraging; however, this may be temporary until feeding needs are satisfied. Additionally, it has been reported that amino acids present in live *Asplanchna* may be recognized as potential food by *Brachionus,* which may also trigger increased swimming speed of *Brachionus* [1]. An alternative response has been described by Zhang et al. [12], where *Brachionus* showed a floating behaviour in the presence of *A. sieboldii*. This behaviour was not found in our study.

Swimming speed and persistence of *A. brightwellii* was constant among treatments. Thus, the predator’s behaviour is independent from the presence or absence of prey. In the field, *Asplanchna* typically faces a number of different prey organisms, ranging from large flagellates over ciliates to rotifers with, potentially, different swimming speeds. This mixture of various prey organisms might explain the unresponsive behaviour of *Asplanchna*. For cruising predators that naturally encounter a variety of prey organisms at the same time, a specific response to a single prey has a low adaptive value.

We found a lower swimming speed (only significant for the strain “Michigan”) and a lower persistence (only significant for strain “IGB”) in the presence of kairomones. This means that the animals were slower and changed their direction more often. Thus, sensing the chemical cue from the predator without physical perception led to a different behaviour than facing the physical predator. Chemical communication is very important, especially for prey with poorly developed eyes. They use chemical signals emitted from other prey individuals and/or predators to evaluate the risk of predation [46]. Their responses to chemical cues often result in a reduced activity level [46,47]. This behavioural response is true for *B. calyciflorus* strain “Michigan”, which decreased its swimming speed in response to predator kairomones. However, *B. calyciflorus* strain “IGB” had no significant reduced response to chemical cues from its predator. This may suggest a strain-specific response to predator cues. In a study of the semi-benthic bdelloid rotifer *Philodina megalotrocha*, an increase in swimming frequency in response to the cue of a copepod predator was found, which might be an escape response of the otherwise benthic prey [35]. *Asplanchna*-conditioned medium might be a complex chemical mixture [48] that could contain many compounds such as residual odours, thus the reactions of strains to these complex chemicals may differ. Preston et al. [1] found *B. calyciflorus* increases its swimming speed in the presence of *Asplanchna*-conditioned medium in contrast to our study. They proposed that *Brachionus* may have recognized the residual odours as food, thus causing an increase in swimming [1,49].

### 4.2. Transgenerational Behavioural Responses with Spined Prey

We found that unlike unspined *Brachionus*, which showed a behavioural response (by increased swimming speed) to live predator presence, spined *Brachionus* showed no increase in swimming speed. This suggests that protected prey individuals are less concerned with predation as compared with their unspined conspecifics. Spined *Brachionus* also exhibited no behavioural response to predator cues, contrary to unspined *Brachionus*, which decreased their swimming speed when exposed to predator cues. This reinforces the assumption that prey individuals with long spines are indifferent towards the presence of predators, which could be attributed to the effective protection provided by the spines. Spines increase handling time, decrease capture rate and can cause damage to the predator´s (*Asplanchna *sp.) body, hence *B. calyciflorus* with spines are less preferred and are sometimes outrightly avoided as opposed to *B. calyciflorus* without spines [16].

Based on morphological and hydrodynamic considerations, we expected a change in swimming behaviour in the presence of spines as found in defended *Daphnia cucullata* [50]. However, we found only an effect of body length on swimming speed but not of spine length. It is well known that body size has an effect on swimming speed [51,52,53]. The absence of an effect of spine length on swimming speed leads to the question of whether the expression of spines is associated with costs in rotifers. Although some studies have reported costs of defence in various species, trade-offs may not arise from a direct allocation cost for formation of defence, but rather from the interaction of the defence with the environment, so-called environmental cost [54]. Other studies have found no consistent trend with fitness parameters and inducible defence in daphnids [55,56,57]. Measuring the costs of inducible defences is quite difficult and has led to contrasting results in the *Brachionus–Asplanchna* predator–prey pair [18]. Using different experimental set-ups, in none out of four studies did spine-induced forms exhibit a clear fitness reduction [58,59,60,61]. Thus, in the cost–benefit relationship, the benefit part is much better understood than the potential costs.

We found no response of the predator to the spined prey, neither in speed, nor in persistence. Thus, the predator’s behaviour was not dependent on the presence or absence of spined prey. As the predator is a generalist, it may not have evolved a specific response to a single prey.

We designed our experiment by exposing the prey to a fixed predation risk. Thus, we cannot make any predictions about the shape of a predation risk–defence relationship. In the field, the predation risk increases with the number of predators and the individual predation risk decreases with an increasing number of conspecifics or alternative prey. These two mechanisms might influence the individual response to predation, in particular the behavioural response. The induced morphological response can also be expressed in a risk-dependent manner, for example, by developing differently sized spines, relative to the perceived risk. In this respect, it would be important to know how the length of the spine affects the mortality of the prey. Further research along these lines would improve our understanding of the behavioural side in predator–prey interactions.

## 5. Conclusions

In summary, the aim of our study was to explore the behavioural response of both spined and unspined *B. calyciflorus* to predation from *A. brightwellii*. Our results with regard to unspined prey revealed a consistent increase in swimming speed for both *Brachionus* strains in predator presence, whereas prey in the kairomones were slower or changed direction frequently. This might facilitate the coexistence of the predator and the prey. We found that unlike non-spined prey, spined *Brachionus* showed no behavioural responses to live predator or predator cues, indicating indifference of protected individuals to predation. This finding suggests that spined individuals behave indifferently to their predator. The mechanism behind this is not yet understood, but it sheds light on a yet unknown aspect of predator–prey interactions and inducible defences.

## Figures and Tables

**Figure 1 biology-11-01217-f001:**
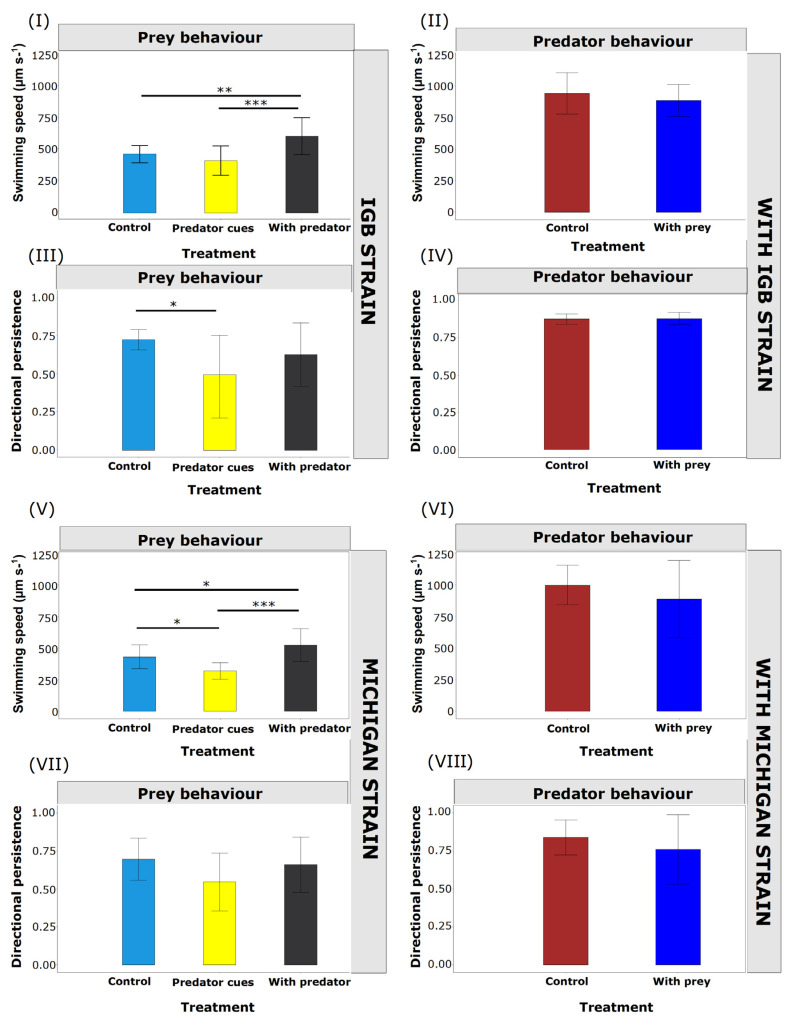
Mean swimming speed (n = 15, μm s^−1^) ± SD and directional persistence (scales from zero to one with values close to 1 indicating that an individual is highly likely to move in the same direction as during the previous time step) ± SD of unspined prey and predators in different treatments. (**I**) unspined *B. calyciflorus* strain “IGB” swimming speed, (**II**) *A. brightwellii* with unspined prey strain “IGB” swimming speed, (**III**) unspined *B. calyciflorus* strain “IGB” directional persistence, (**IV**) *A. brightwellii* with unspined prey strain “IGB” directional persistence, (**V**) unspined *B. calyciflorus* strain “Michigan” swimming speed, (**VI**) *A. brightwellii* with unspined prey strain “Michigan” swimming speed, (**VII**) unspined *B. calyciflorus* strain “Michigan” directional persistence, (**VIII**) *A. brightwellii* with unspined prey strain “Michigan” directional persistence. *p*-value < 0.001 (***), *p*-value < 0.01 (**) and *p*-value value < 0.05 * indicates significance. No asterisk denotes no significant difference between treatments.

**Figure 2 biology-11-01217-f002:**
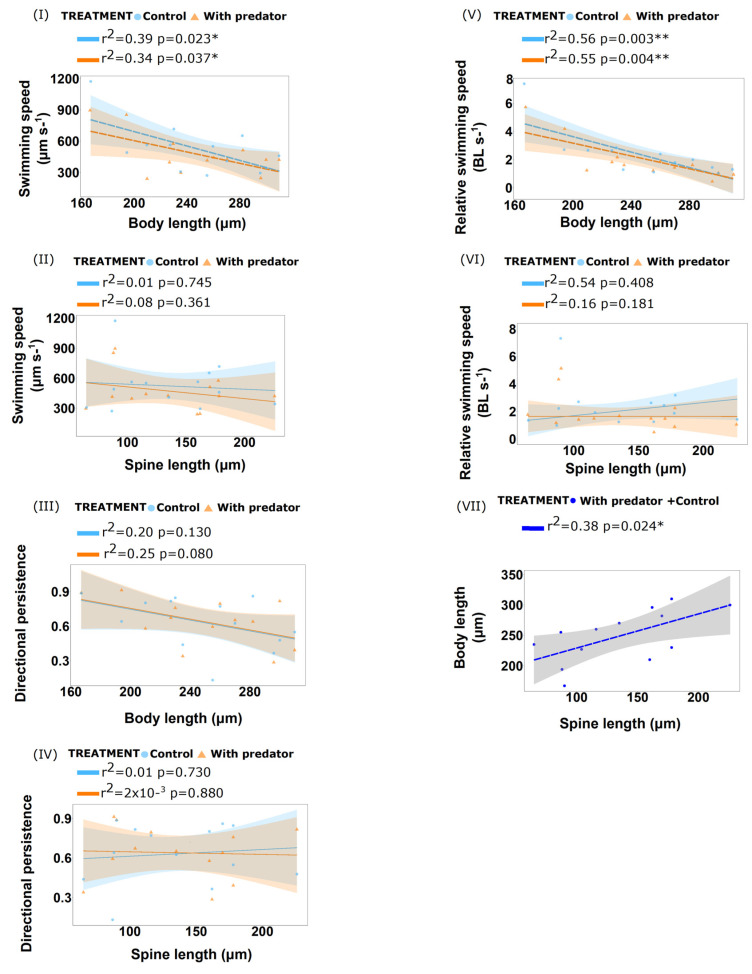
Linear regression plots showing behavioural responses of spined *B. calyciflorus* strain “Michigan” to Live Predator treatment. (**I**) Relationship between swimming speed (µm s^−1^) vs. body length (µm). (**II**) Relationship between swimming speed (µm s^−1^) vs. spine length (µm). (**III**) Relationship between directional persistence vs. body length (µm). (**IV**) Relationship between directional persistence vs. spine length (µm). (**V**) Relationship between relative swimming speed (BL s^−1^) vs. body length (µm). (**VI**) Relationship between relative swimming speed (BL s^−1^) vs. spine length (µm). (**VII**) Relationship between body length (µm) and spine length (µm). Bold dashed lines (**- - -**) indicate significate regressions. Shaded part denotes the 95% confidence interval (CI). *p*-value < 0.01 (**) and *p*-value < 0.05 * indicates significance.

**Figure 3 biology-11-01217-f003:**
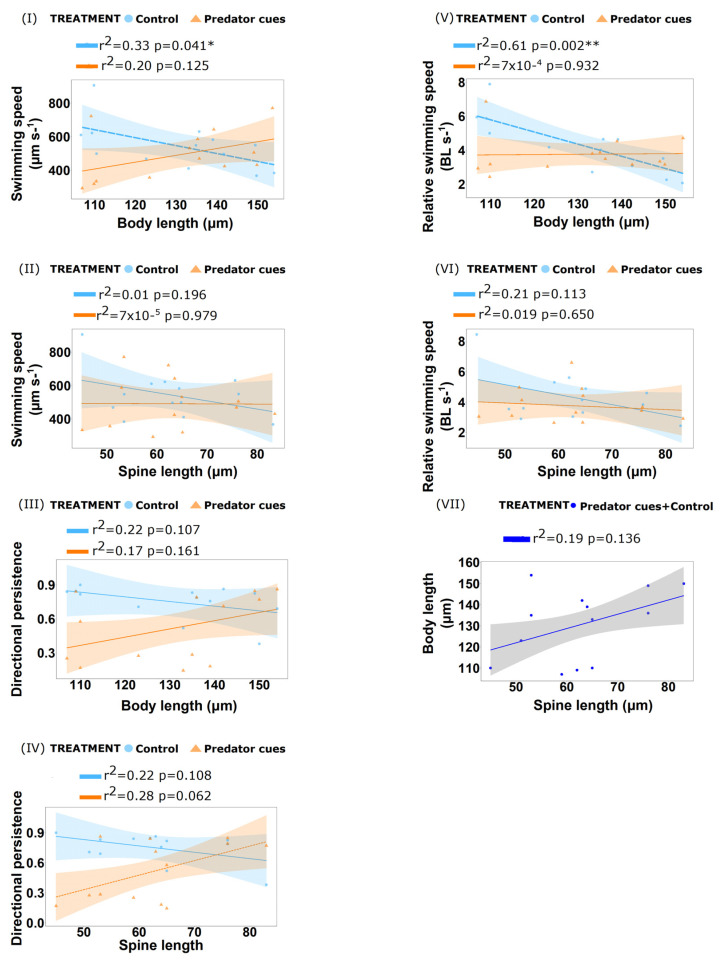
Linear regression plots showing behavioural responses of spined *B. calyciflorus* strain “Michigan” to Predator cues (Kairomones) treatment. (**I**) Relationship between swimming speed (µm s^−1^) and body length (µm). (**II**) Relationship between swimming speed (µm s^−1^) and spine length (µm). (**III**) Relationship between directional persistence and body length (µm). (**IV**) Relationship between directional persistence and spine length (µm). (**V**) Relationship between relative swimming speed (BL s^−1^) and body length (µm). (**VI**) Relationship between relative swimming speed (BL s^−1^) and spine length (µm). (**VII**) Relationship between body length (µm) and spine length (µm). Bold dashed lines (**- - -**) indicate significate regressions. Shaded part denotes the 95% confidence interval (CI). *p*-value < 0.01 (**) and *p*-value < 0.05 * indicates significance.

**Table 1 biology-11-01217-t001:** ANCOVA analyses of swimming speed (μm s^−1^) of spined *B. calyciflorus* strain “Michigan” with different spine lengths and body lengths in laboratory experiments, *p*-value < 0.001 (***) and *p* value < 0.05 * indicates significance.

Swimming Speed (µm s^−1^)					
Treatment	Strain	Variable	df	F	*p*-Value
Live predator	*Brachionus calyciflorus*—“Michigan”	Treatment	1	0.58	0.456
		Body length (µm)	1	15.84	<0.001 ***
		Spine length (µm)	1	2.43	0.137
		Treatment × Body length	1	0.89	0.359
		Treatment × Spine length	1	0.88	0.359
		Body length × Spine length	1	3.75	0.069
		Treatment × Body length × Spine length	1	0.70	0.413
Predator cues (Kairomones)	*Brachionus calyciflorus*—“Michigan”	Treatment	1	1.22	0.284
		Body length (µm)	1	0.08	0.781
		Spine length (µm)	1	1.23	0.282
		Treatment × Body length	1	7.84	0.012 *
		Treatment × Spine length	1	0.08	0.782
		Body length × Spine length	1	0.02	0.890
		Treatment × Body length × Spine length	1	6.28	0.022 *

**Table 2 biology-11-01217-t002:** ANCOVA analyses of relative swimming (BL s^−1^) of spined *B. calyciflorus* strain “Michigan” with different spine lengths and body lengths in laboratory experiments, *p*-value < 0.001 (***) and *p*-value < 0.05 * indicates significance.

Relative Swimming Speed (BL s^−1^)					
Treatment	Strain	Variable	df	F	*p*-Value
Live predator	*Brachionus calyciflorus*—“Michigan”	Treatment	1	0.46	0.506
		Body length (µm)	1	31.53	<0.001 ***
		Spine length (µm)	1	2.16	0.159
		Treatment × Body length	1	0.95	0.342
		Treatment × Spine length	1	0.78	0.389
		Body length × Spine length	1	7.49	0.014 *
		Treatment × Body length × Spine length	1	0.39	0.543
Predator cues (Kairomones)	*Brachionus calyciflorus*—“Michigan”	Treatment	1	1.89	0.186
		Body length (µm)	1	4.50	0.048 *
		Spine length (µm)	1	0.89	0.357
		Treatment × Body length	1	6.99	0.017 *
		Treatment × Spine length	1	1.7 × 10^−3^	0.968
		Body length × Spine length	1	0.02	0.899
		Treatment × Body length × Spine length	1	6.29	0.022 *

**Table 3 biology-11-01217-t003:** ANCOVA analyses of directional persistence of spined *B. calyciflorus* strain “Michigan” with different spine lengths and body lengths in laboratory experiments, *p*-value < 0.01 (**) and *p*-value < 0.05 * indicates significance.

Directional Persistence					
Treatment	Strain	Variable	df	F	*p*-Value
Live predator	*Brachionus calyciflorus*—“Michigan”	Treatment	1	0.01	0.909
		Body length (µm)	1	12.38	0.002 **
		Spine length (µm)	1	3.42	0.054
		Treatment × Body length	1	0.12	0.731
		Treatment × Spine length	1	0.31	0.582
		Body length × Spine length	1	0.05	0.823
		Treatment × Body length × Spine length	1	0.89	0.358
Predator cues (Kairomones)	*Brachionus calyciflorus*—“Michigan”	Treatment	1	7.55	0.013 *
		Body length (µm)	1	0.03	0.856
		Spine length (µm)	1	0.67	0.425
		Treatment × Body length	1	1.35	0.260
		Treatment × Spine length	1	3.19	0.091
		Body length × Spine length	1	0.78	0.390
		Treatment × Body length × Spine length	1	0.21	0.651

## Data Availability

The data presented in this study are openly available in repository FigShare at https://doi.org/10.6084/m9.figshare.20141546.v1.

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
