# Peer review of "Behavioural Responses of Defended and Undefended Prey to Their Predator—A Case Study of Rotifera"

_biology, 2022, doi:10.3390/biology11081217_

Round 1

Reviewer 1 Report

The Authors investigated behavioural responses of a prey and its predator to each other’s presence, or chemical cues thereof, in a model rotifer-rotifer two-species system. They asked an important question whether morphologically defended prey responds differently than undefended prey. Overall, the study was designed well and executed properly. The aim of the study and the hypotheses are expressed clearly and follow logically the introduction. Also, the results are generally discussed properly. However, the text needs thorough editing, including introducing more language rigour in the introduction, completing method description, editing results presentation and discussion.

1. Information lacking in method description:

Can the Authors provide more details from what pond Asplanchna came from? (l. 94-95)

For the experiment, where the animals taken randomly from the culture and where different treatments executed simultaneously?

How long was the transgenerational induction before experiments? (l. 106)

What was the animal density in the cultures? I expect this can affect the studied responses.

Were there any kills in the predator-presence experiments?

How was the automatically obtained video data curated? Was the automated reading flawless?

Was linear regression modelling used (l. 206)? – later it appears it was.

How was the statistical analysis performed? What software was used? What functions, Anova type etc.?

2. Issues with results presentation:

Asterisks (or horizontal lines or using other method) in Figure 1 should clearly mark which treatments differ and include all the information from Table 1, which then should be omitted.

In Figure 2, please mark significant and no-significant regressions differently, e.g. dashed versus solid lines or by differentiating clearly line width. The graphical information is misleading now as pointing to potentially non-existent relationships. Also, unifying scales between columns 1 and 3 and between 2 and 4 will be helpful.

Language rigour: lines 247-248, 262-263 and elsewhere: the Authors found no differences between treatment and control, not “no change”. Change could be found only where the same individuals were recorded before and after treatment application.

Lines 214-216 need to go out. Line 221: “our (?) treatment”. Line 228: is M strain meant here? Tables 2 and 3 are referred in the text before Figure 2 – please, change order of appearance.

3. The introduction does not read well and needs both substantial and language thorough editing. While this chapter has a logic flow, its language needs more precision. Beginning with the first sentence as an example: “Every individual species” (30) – individuals, i.e. individual organisms, process information rather than whole species, and I understand this should read “Every individual” or “every organism”. Then, “information conveyed in its environment” does not necessarily reach the organism – it seems “conveyed in” does not convey what is meant here, I suggest using a different phrase. Further on, neither “information can serve as opportunity” (31-32) nor “predation is a life process” (33) is a good phrase. Yet another example of many: “sensing predators” does not include any “defences” (44-46). Jumping on, escape more often is used upon attack, not mere detection (40-41, 47-48), and morphological defence does its work when the prey is captured, not merely attacked (68-69). One could go on doubting over almost every sentence. I strongly suggest consulting the text with an English native professional, biologist at least.

4. Discussion:

Increased activity is not so surprising in the presence of cruising or actively hunting predators (e.g. Parigi et al. 2019). Also, increased swimming in a rotifer in the presences of predator chemical cues has been recently reported by Parysek and Pietrzak (2021), the Authors may want to refer to that.

Unlike most rotifers, Asplanchna is not a filter feeder (l. 345). Please, explain or correct.

The logic is unclear with chapter 4.2 on transgenerational responses beginning with body size relevance (l. 364-365).

What is “the second line” of defence? (l. 369). I do not understand “this allowed” (l. 374). Also, lines 393-394 are hardly comprehensible. There are further language edits needed in the discussion, revision by a native speaker is suggested.

Reviewer 2 Report

Reviewer report of the manuscript Biology-1799513

In summary, this is a well-written manuscript highlighting the behavioral responses in the presence (or absence) of an additional morphological line of defense of prey to a presence of predators or their chemical signatures. 

While the current version of the manuscript stands quite strong, several limitations makes me not recommend it to be accepted in the present form or as a minor revision. These limitations should be addressed either by (i). provding additional and clear details about the methodology, (ii). performing additional analyses, or (iii). providing strong justifications if the authors decide not to perform the above additional analyses.  

In my chapter-specific comments below, the text highlighted in bold red are the main aspects that the authors need to be focused on and invest considerable time. 

I commend the authors for performing this intersting work and wish them all the best with the revision, which would make this manuscript further strong. 

INTRODUCTION

Line 34: Please change to “may die” (predator encounters do not always result in death. See Lines 69-70 for self-explanation)

Line 48: Is the “crustaceans” required here? In this paragraph, the general emphasis is on zooplankton – and DVM is not exclusive to crustaceans (may, be the cited material is crustacea-specific. In that case, cite more of a generic review/overview – which are plenty).

Line 50: Two other somewhat less-studied potential anti-predator behaviors of zooplankton are swarming and schooling. It would be nice to include those. Empirical evidence for these is mostly marine (browse for literature in recent times for a better overview).

Line 59: “strategies and techniques”: what is the difference between the two? Based on how the sentence is written, is a survival technique different from a survival strategy? How?

Line 62-64: Excellent point!

Line 74: “as there is no orientation from a distance”: do you mean to say, “as there is no sense of orientation from a distance”? The current writing is a bit unclear. Consider rephrasing.

Line 77: “when they live together” sounds more like spoken English (could be a personal preference). Consider using “in coexistence” or something likewise.

Line 79: “predation risks”: do you mean to say, “perceived predation risk”?

Line 81-82: “Analysed” and “analysis” appear in two places in this sentence leading to the question that ´did you analyse the analysis´? Use an alternative term for the “analysis” at line 82 (“videographic techniques”, for example).

Line 83: Change to “accurate details”

MATERIALS AND METHODS

Line 98: I happen to remember “sensu stricto” as the correct term. Please check and adjust accordingly.

Line 94-96: It appears that the predator in captivity was fed a mixed diet (two strains of Brachionus and Keratella). Was there any specific reason to use Keratella? Was it due to the prey composition in the predator´s natural habitat?

Generic remark: If the journal allows, it would be nice if you can add an image of the two strains “Michigan” and “IGB” to the methods. Visible scale morphological differences should be expressed visually – so the manuscript is more appealing for a generic and wider (sometimes even non-scientific) audience. Search engine image searches will also pick-up your embedded images and hence increasing the potential viewership of the manuscript (& citations).

Line 108-109: Is this predator size temperature- and food-driven or did you artificially select for this size?

Line 116: “macro lens” be specific.

Line 118: This reads like there was another camera for video recording. Dial down the statement, for example, “using the above camera” or something likewise.

Section 2.2: Add some information about the container (petri dish, counting cell, plate, cube or whatever) that the animals were put to record their movements. Provide dimensions of the container if possible – so the reader has an idea about the size of the environment compared to the size of the animals (which can be a key limitation for movements). [This remark also ties with commments further below...]

Lines 124 & 125: Since these sizes are study-specific, use “ranged”. Predator size =  repeated information (see Line 108-109).

Line 126-131: “exposed to A. brightwellii” ´: does this mean exposure prior to video recording or simultaneous exposure and video recording?  - or both? If possible, please be specific. (The explanation is at Line 136 onwards, but it is good to be specific at the front if possible).

 Line 134: “control treatment”: just say, “control”

Line 141-142: So, were there any cases where the predator killed or harmed the prey, particularly given that it was starved? Was the behavior of all five prey individuals recorded or only one (or more) were picked and recorded?

In case of the former, how did you tackle the potential motility changes caused by predator attacks and the ensued physical trauma?

In case of the latter, how did u pick specific individuals without/minimizing bias?

Line 144: How dense? Do you have any estimate of their abundance? (it would be helpful to include such and OK if you do not have an abundance estimate).

Line 178: “four locations per second”: what is meant by "locations"?

Line 180: Once again, these findings are dependent on the container dimensions. Therefore, please provide those as stated above. Note that even if the container dimensions appear to limit the movement of the individuals, you may justify your methods if other studies have been conducted using a similar setup.

Line 185: Superscript the “TM”

Line 185: Is BEMOVI not reliant on the GPU of the system more than the CPU? Check and if so, provide the GPU information alongside the CPU and RAM specifications. Also, it says, “9 64 based processor”. What does “9” mean?

RESULTS

Line 214-216: Remove this part.

Line 220: The earlier question comes back: was this mean swimming speed (say, V) a mean calculated across the filming duration (Vt). Or mean calculated both of filming duration and per-individual (Vt,i assuming > 1 individuals were filmed at the same time)? Or mean calculated across time and/or individual and the replicates (Vt,R or Vt,i,R)?

Figure 1 (& elsewhere necessary): Why did you prefer to use SEM instead of SD? Was it due to the velocities were extracted from a trajectory tracking? The use of Mean ± SD would also have been nicer because it shows the amount of variability around the mean (as prey individuals encountering predators, esp. in a potentially physically-constrained habitat move erratically and their movement velocities change rapidly across time). Justify your choice of the used dispersion metric. If you decide to stay with SEM, then, give the reader some sense of the sample size (on Table 1 for example), so they can back-calculate the SD from SEM. This is another reason why the comment above on Line 220 is important (as the sample size varies depending on how the mean velocity was estimated).

Also, Panel II is not labelled on the figure. Also, add the vertical labelling tab (“IGB”, “MICHIGAN”) to the left panels (I & III), as only the prey had two strains – predators came from the same pond!

Also, you could have optimally presented this data using a series of boxplots rather than barplots. Nonetheless, it is just my personal preference, and it is ultimately up to the authors to decide the type of presentation depending on their preference.

Table 1: If the asterisks marked on the barplots are significance indicators extracted from this table, they do not seem to match. For example, there is no p < 0.001 scenario at the IGB panel (I). Check this and make adjustments accordingly.

Also, if the significant codes can be adequately presented on Figure 1 (using asterisks), consider moving this table to the Appendix/Supplement.

Line 265 & 267 & Figure 2 (b) (I): Why is the coefficient of determination (R2 = 0.33) produces two considerably different p-values (one statistically significant at 0.05 and other not) under the same degrees of freedom?

DISCUSSION

Line 325: Did you try comparing, first, the size of the container during filming across your study and the cited references nos. 10 and 41? Can it be that the escape routes of the prey were cut due to physical constraints of the container it was put in during filming? For a zooplankter occupying a natural habitat (say, a rotifer in a shallow pond), potential escape routes are three-dimensional. Did the container used in your study during filming allow the behavior of prey to exercise their escape across all the three dimensions (including the vertical dimension)? This is why I requested to provide the type and size of the container used to put these organisms in the filming process in the Methods chapter. Further, assuming that you provided adequate space and dimensionality for the escape behavior to take place, is swimming speed the only proxy of prey´s escape response? What about the changes in swimming direction (trajectory)? Rapid changes in the swimming directions are oftentimes used by prey individuals outmaneuver a predator. Since your motion analysis has a trajectory analysis functions, why didn’t you use it? If you decide to stay with swimming speed as the only proxy of behavior, I wish to see a strong defense for that choice in the discussion with references to related literature, which solely relied on the same proxy.

Also, did you compare the duration of the experiment across these studies and yours? If your filming duration (30 s) is significantly shorter than those of the cited studies, then the discrepancy is self-explanatory. Swimming speed of a motile zooplankter enclosed in space-limited habitat will be higher in the beginning and will decrease with time as the animal is exhausted and there are no escape routes.

Line 357: Yes, it could well be a strain-specific response. Can the underpinning cue for such strain-specific response be the concentration of the kairomone (detection threshold and/or reaction threshold)? Is there any literature about this? This is why I asked in the methods chapter “how dense?” when you mentioned that kairomones were taken from a dense aggregate of the predator.

This leads to another remark. I feel that a key limitation of your study is that it is based on the prey response of a fixed (or prescribed) level of predation risk. This is apparent in the kairomone concentration as well as the live presence of the predator. A more interesting question to be asked would have been, “what is the expected prey behavioral response when the predator numbers (density) increases/kairomone concentration increases?” I am absolutely not recommending to do more experiments, but this should be tackled in a future experiment and more importantly be mentioned as a limitation & a future extension of the current approach.

Line 378-379: “less concerned with predation”: do your results support this argument? Is “spined ones are as concerned as the non-spined ones” the safer argument to make given your findings? Correct me if I am wrong.

Line 387-389: This is another reason why I argued above that mean swimming speed is not the only (and perhaps not the ideal) proxy of escape behavior of prey zooplankton. A proper trajectory analysis may shed new light on your findings if done properly. Is there a way that such an analysis can be performed within the scope of this study, at least as a supplementary material?

Line 399: “From the predator´s side”: Reads like spoken English. Use “perspective” in place of “side” or something likewise.

General remark: In a discussion, you need to look at your manuscript in hindsight and need to be a bit self-critical about the limitations of your approach/methods/data/analyses. At least for me, “these are our findings and conclusions, given these limitations” sounds more scientifically appealing than “these are our findings and conclusions”. I am not suggesting adding a paragraph describing potential limitations, but you could use statements wherever you feel the need of critically looking at your findings in the existing text (e.g., “our findings suggest blah blah blah[…..]However,……..”). 

Reviewer 3 Report

Review of: Behavioural responses of defended and undefended prey to their predator

By: Victor Parry, Ulrike E. Schlagel, Ralph Tiedemann, and Guntram Weithoff

Summary:

This study explores the relationship between induced morphological response and behavioral response in the presence of a predator. The novel finding that this study attempts to explore is how presence of spines, an attribute that presumably reduces predation, allows an individual to behave as if there were no predator despite the presence of risk to predation. This study uses speed of prey as an indicator for response to predation.

General comments:

1. The present study poses an interesting question and presents data that are tantalizing and, yet overlooks the key aspects; speed is not the only indicator that an organism is responding to potential predation. I will try to use an analogy to demonstrate the point, which should be addressed. When a person encounters a known predator, like a tiger, some people may flee and some may freeze. In either case we would describe both as responding to the predator. It is possible that the large spined individuals are exhibiting some important defensive behavior that is not detectable by measuring speed. To help better explore this relationship I would suggest one of two avenues, though both together would be ideal. First, movement data from the video clips be used more fully to explore whether a different behaviour type is exhibited. One option being used in similar mesocosm landscape of fear studies with fish is the use of a multivariate approach. This would allow for a clearer understanding of different response types of the various treatments (I suggest considering visualizing with an NMDS and testing with a PERMANOVA). Second, the authors can demonstrate that predation risk is reduced proportionate to the change in speed for spineless individuals and spined individuals are impervious to predation independent of speed; it is logical that if bigger spines cause a reduction in predation risk behaviour then predation on large spined individuals should be equal or lower than small spined individuals in the presence of a predator. This is assumed in the text, but not evaluated though the authors put a great deal of weight into stating that they are examining this response for fitness reasons. Either of these two avenues would allow the reader to make an important conclusion, either 1) behaviour type changes concurrently with morphology or 2) risk to predation changes with morphology justifying changes in behaviour.

2. Environmental conditions – the arena where the trials take place is poorly described and may be suppressing or enhancing a signal. What precedent is there for trials in an arena like this? Lighting, for example, has been shown to strongly affect behaviour of zooplankton, might this affect the results? Were pilot experiments conducted that showed that all wells on the plate had similar conditions and similar responses – specifically I am wondering whether Snell’s window and movement in the lab was considered. Finally, how was arena size chosen? Are densities within the arena related to what occurs in nature or are they simply what was convenient for study, in which case it might be good to supplement this data with a supplemental materials component indicating that the density and arena size were not influential in the behaviour. My concern here is that behavioural responses can be informed by the environment in which they take place.

Minor comments:

There are several areas that should be addressed that would greatly improve the manuscript.

The paper title seems to greatly exceed the scope of the experiment – The paper focuses on speed changes of two different strains of the same species of rotifer. Do Daphnia exhibit the same signal?

Strain background – Providing background on the strains and why they were chosen would be useful. Are these a model pair of strains that allow for the testing of this hypothesis? Is their background work on their general behaviours that would help in understanding the different responses?

Related to the above – The scope of much of the writing needs to reflect the scope of the study and the data collected. Specifically, the present study focuses on two strains of the same species; would other zooplankton exhibit the same or different patterns?

Timing – Similar to the above comment on how the environment was chosen, what precedent is there for the timing of the acclimation period and trial period? In studies of both fish and Daphnia the literature indicates that timing is an important feature.

Predator-Prey ratio – A 1:1 ratio is described. This is a high predator to prey ratio, why was this chosen?

Specific line comments:

Because of the need to address the major and minor revisions described above, only a few select line by line revisions are provided to indicate areas that could be improved upon.

Line 22: The phrasing ‘“aware” of their defence’ is inappropriate as that is not what was tested, as is the framing of this sentence. This should be written such that it more closely aligns with the scope of the study and what was measured. It’s possible that they are “aware” of their defense, or it could be that they exhibit a different style of defensive behaviour.

Introduction: A great deal of emphasis is placed on organismal interactions and predator-prey interactions and little on kairomones and the role they play in biology and ecology. These should be reversed because the former is well established and the latter is more important to the niche this manuscript is attempting to fill.

Lines 40-41: The phrase “increase the chance of escape when detected” does not make sense as its own item in this list. It is another part of the “ability to” mechanism listed before it. Might be clearer written as “This can include the ability to avoid detection by their predator (camouflage and crypsis), the ability to recognize and increase detection of approaching predators, as well as increase the chance of escape when detected, and antipredator morphological defences…”.

Lines 41-42: The commas in the phrase “antipredator morphological defences which can be permanently occurring, or induced by chemical cues released by the predators” obscure its meaning. Would be clearer with a comma before “which” and no comma before “or”.

Lines 47-48: No comma needed before “when”. Same as in following instances (e.g., Line 77).

Line 49: Was the item supposed to be “change in speed due to predator presence”? Missing similarly in other following instances.

Line 52: The transition to discussing morphological defenses feels abrupt; it is unclear whether this is supposed to be a new point from the behavioral changes. A transition sentence would clarify this.

Line 69: The point might be easier to follow if a new sentence was started at “however”. “However” should also be followed by a comma in either case. Same as in following instances (e.g., line 79).

Line 74: What does “no orientation from a distance” mean? (Is it just referring to max. radius of chemo-/mechanoreceptors? Or does orientation refer to something other than changed behavior?)

Line 76: No comma needed after “both”.

Line 132-133: Should it be “five non-egg-bearing individuals were randomly chosen”? Because the emphasis is on the random selection; the fact that they were non-egg-bearing was non-random (done intentionally to ensure comparable results across replicates), as I understand it.

Results section: Although not necessarily incorrect, some sub-sections it was difficult to figure out what was being described. Because figures and tables also present material, one way to improve this is to shift the descriptions into more prosaic language that is easier to follow.

Round 2

Reviewer 2 Report

I thank the authors for considering the comments by all reviewers in adjusting and improving the manuscript. I am satisfied with the changes made to the manuscript during the revision. While no further comments exist at this point, I believe that the manuscript can be accepted for publication at its current form.

I wish the authors all the best in their future research efforts.

Author Response

Thank you very much for reviewing our manuscript